# Kinetic Effects of Ciprofloxacin, Carbamazepine, and Bisphenol on Biomass in Membrane Bioreactor System at Low Temperatures to Treat Urban Wastewater

**DOI:** 10.3390/membranes13040419

**Published:** 2023-04-07

**Authors:** Laura Antiñolo Bermúdez, Antonio Martín-Luis, Juan Carlos Leyva Díaz, María del Mar Muñío Martínez, José Manuel Poyatos Capilla

**Affiliations:** 1Department of Civil Engineering, Institute of Water Research, University of Granada, 18071 Granada, Spain; 2Department of Chemical Engineering, Institute of Water Research, University of Granada, 18071 Granada, Spain

**Keywords:** wastewater treatment, bisphenol A, carbamazepine, ciprofloxacin, temperature effect, membrane bioreactor, respirometric test

## Abstract

This study analysed the kinetic results in the presence and absence of micropollutants (bisphenol A, carbamazepine, ciprofloxacin, and the mixture of the three compounds) obtained with respirometric tests with mixed liquor and heterotrophic biomass in a membrane bioreactor (MBR) working for two different hydraulic retention times (12–18 h) and under low-temperature conditions (5–8 °C). Independently of the temperature, the organic substrate was biodegraded faster over a longer hydraulic retention time (HRT) with similar doping, which was probably due to the longer contact time between the substrate and microorganisms within the bioreactor. However, low values of temperature negatively affected the net heterotrophic biomass growth rate, with reductions from 35.03 to 43.66% in phase 1 (12 h HRT) and from 37.18 to 42.77% in phase 2 (18 h HRT). The combined effect of the pharmaceuticals did not worsen the biomass yield compared with the effects caused individually.

## 1. Introduction

Today, society has a strong environmental conscience, and attempts are being made to minimise the impact of human activities on the environment [1]. For a resource as important to the population and human activities as water, the correct management and treatment of wastewater is fundamental. Inadequate treatment can release contaminants into the natural environment. Over the last few decades, there has been a rise in the number of emerging contaminants detected in water around the world [2]. These pollutants are pharmaceutically active compounds and endocrine-disrupting chemicals [3]. Pharmaceuticals may induce a variety of physiological changes, which can be reversible or not, in non-target aquatic organisms [4]. They can reach the aquatic environment via a high number of routes, such as discharge of both raw and wastewater treated by municipal [5], hospital [6], and industrial wastewater treatment plants (WWTPs); sewer leakage [7]; landfill leachate [8]; or treated wastewater used for irrigating [9]. These compounds can be found in both raw influents and treated effluents of WWTPs, so they are of particular interest [10]. However, existing treatment plants are not designed to remove such contaminants, leading to discharge into aquatic environments [11].

Within this group of contaminants are carbamazepine, bisphenol, and ciprofloxacin. These pharmaceuticals have been chosen for the study because of their widespread use worldwide, their detection in wastewater plants, and their presence on the European Union list of substances for monitoring purposes at the European level.

Bisphenol A (2,2-(4,4-dihydroxydiphenyl) propane) is an endocrine-disrupting compound (EDC), which can alter the functioning of the endocrine system, with adverse effects on both humans and the environment. Among all EDCs, bisphenol A has more detrimental effects than others [12]. It can cause sexual dysfunctions; act as a carcinogenic agent; affect neuroendocrine functions; and increase the risk of certain diseases, such as diabetes [13,14].

Carbamazepine is an anti-epileptic drug. Its consumption is about 1000 tons per year in the world [15] and can be found in a large number of water systems [16]. Among its side effects, it has been shown that exposure to carbamazepine has negative effects on both the central nervous and digestive systems, as well as on the development of embryonic and blood cells [17].

Ciprofloxacin is an effective antibiotic when it comes to treating some kinds of diseases caused by bacillus bacteria, some kinds of Gram-positive bacteria, and entero-bacteriaceae. It has been detected in higher concentrations than other antibiotics due to its high stability against degradation because of the presence of a fluorine ion in its structure [18]. Its spread in the environment can cause antibiotic resistance, resulting in a potential long-term threat to humans [19].

Membrane bioreactor (MBR) systems have become an advancement over conventional activated sludge systems for wastewater treatment by making higher-quality effluents possible, being able to work at a high MLSS concentration, and allowing a reduction in space requirements to be achieved [20,21]. This technology also has a higher removal rate of emerging pollutants [22,23], and its capital cost is the most relevant disadvantage, which could be reduced with standardisation [24].

It is important to research not only the removal rate of those compounds but also the effect that they can exert on organic matter removal by heterotrophic biomass. To the best of our knowledge, there are no studies in the literature that report on the influence of any of these three compounds on the heterotrophic biomass performance of an MBR system at low temperatures. Other studies have analysed the kinetics and biodegradability of bisphenol A (BPA) in biological systems [25], oxidation processes [26], its removal in a nitrifying system with immobilized biomass [27], and biomass performance at higher temperatures [28]. Regarding carbamazepine, there are studies that report its removal in an MBR system [29] and other systems [30,31]. Lastly, regarding ciprofloxacin, there are studies that report its removal with different types of WWTPs [32,33] and thermal procedures [34], but none of those studies are focused on the kinetic effects at low temperatures.

Among all the variables that affect wastewater treatment, temperature has turned out to be the one with the major influence [35,36], and many WWTPs are exposed to high seasonal temperature variations.

The aims of this study were to study the effects of emerging pollutants (Bisphenol A, carbamazepine, and ciprofloxacin) on biomass stability for the correct operation of the membrane bioreactor system at low temperatures, to study the capacity of the membrane bioreactor to degrade organic matter at low temperatures, and to carry out a kinetic study in the absence and presence of emerging pollutants on samples from the above bioreactor.

## 2. Materials and Methods

### 2.1. Pilot Plant

Samples were taken from a pilot plant located in the Los Vados WWTP in Granada, Spain. It consists of two main parts: an aerated cylindrical bioreactor that functions as a mixing tank for the recirculation flow and the influent, and an 84 L tank with four submerged hollow-fibre ultrafiltration membrane modules attached (ZW-10; ZENON^®^), whose pore size is 0.04 µm and whose area is 0.93 m^2^ per membrane. Municipal wastewater from the primary settling tank of the Los Vados WWTP is the influent of the bioreactor. The entire system is schematically represented in Figure 1.

The plant operates with combined cycles of filtration and backwashing with periods of 9.67 min and 0.33 min, respectively. Filtration is carried out from the outside to the inside by means of sucking. A recirculation pump from the membrane tank to the mixing tank allows a uniform MLSS concentration to be maintained. The membrane bioreactor is supplied with a constant air flux, and waste sludge is purged from the system.

The study was carried out during the months of December and January. Under the plant operating conditions of 12 HRT (flow rate of 7.02 L/h) and 18 HRT (flow rate of 4.67 L/h) and at low operating temperatures due to the climatic conditions of the region, no increase in the fouling of the membrane modules and transmembrane pressure was observed. A cleaning and recovery cycle of the membrane modules was not necessary. The membranes worked at low fluxes and low permeability. Table 1 shows the working permeability values, which were maintained at 1.29–2.22 L/(m^2^ h bar).

For the characterization of the pilot plant, influent and effluent samples were analysed (temperature, pH, conductivity, MLSSs, MLVSSs, BOD_5_, and COD). The results obtained are reported in Table 2.

In the effluent, values of suspended solids surpassing 5 mg/L were detected in some cases, which may have been due to the membrane backwash from the effluent tank during the different experimental phases.

### 2.2. Reagents

BPA was obtained from Sigma-Aldrich Co. (St. Louis, MO, USA). A solution of BPA (97%; CAS No. 80-05-7; MW: 228.29 g/mol) was prepared by dissolving 2 mg of the compound [28] in 2 mL of HPLC-grade methanol (Merck, Darmstadt, Germany) [25,37] and added to the respirometer during the homogenization phase of the doping assay. Moreover, 2 mL of methanol was added to the respirometer during the control homogenization assay to avoid side effects generated by methanol addition in the doping test.

A total of 1 mg of carbamazepine was added during the doping phase to reach a concentration of 1000 µg/L in the respirometer, and 0.1 mg of ciprofloxacin was added in the same way to maintain a concentration of 100 µg/L [38].

Sodium acetate was obtained from AppliChem GmbH (Darmstadt, Germany). A stock solution of sodium acetate (99%; CAS No. 127-09-3; MW: 82.03 g/mol) of 500 mg/L was prepared; then, three different dilutions of 35, 70, and 100% [39] were used in the assays.

Mixture assays were carried out by adding the three above compounds at concentrations identical to those in the single-doping test.

### 2.3. Respirometric Assays

Eight different pairs of control and doping respirometric experiments were carried out. Each test of the pair had a homogenization phase, consisting in reaching an oxygen saturation state in the sample; an exogenous phase, in which the dynamic oxygen uptake rate was measured under stirring, recirculation, and aeration conditions; and an endogenous respirometric assay, which was performed to evaluate the decay coefficient by leaving the mixed liquor without aeration so that the dissolved oxygen concentration decreased to zero. They were carried out in two different phases, varying the HRT from 12 to 18 h, as reported in Table 3.

It can be seen that under the operating conditions, the F/M ratios corresponded to low organic loads with respect to the concentration of microorganisms to compensate for the low-temperature effect.

Respirometric tests were carried out using a flowing gas/static liquid-type batch respirometer called BM-Advance. The respirometer worked at 20.0 ± 0.1 °C temperature, 7.25 ± 0.50 pH, 0.906 ± 0.001 L/min air flow rate, and 2000 rpm stirring rate, and recirculation was set up to assure homogenization in the respirometer. Mixed liquor samples of one litre were transferred from the pilot plant to the respirometer for each of the assay pairs.

Firstly, each sample was homogenized and aerated for 24 h before the assay started [39]. Then, the first exogenous control respirometric assay was carried out by adding the three known sodium acetate dilutions S_1_, S_2_, and S_3_, as shown in Figure 2.

Afterwards, an endogenous respiration assay without air flow was carried out, resulting in a graph similar to that in Figure 3.

The R_s_ rate is the rate of consumption in exogenous assays in the presence of a substrate and a constant supply of O_2_, while the OUR rate is the rate of consumption in the absence of a substrate and without O_2_ supply.

Finally, 2 identical assays were carried out after doping the pertinent compounds into the same sludge sample.

### 2.4. Kinetic Modelling

In order to model the kinetics of the analysed biomass, the yield coefficient of heterotrophic biomass referred to oxygen (Y_H_), the maximum specific growth rate of heterotrophic biomass (µ_max_), the half-saturation coefficient of organic matter (K_M_), and the decay coefficient of heterotrophic biomass (b_H_) were calculated following the kinetic parameter estimation of heterotrophic biomass [28]. This estimation consisted of eight different steps.

Step 1: By integrating the R_s_ from Equation (1), oxygen consumption (OC) was determined for each of the three additions of sodium acetate dilutions (S_1_, S_2_, and S_3_).
(1)$$\mathrm{OC}=\int_{t_{0}}^{t}R_{s}{\mathrm{dt}\;(\mathrm{mgO}}_{2}/L)$$

Step 2: Y_H_ was calculated with Equation (2).
(2)$$Y_{H}=\frac{S-\mathrm{OC}}{{S\cdot f}_{\mathrm{cv}}}\;(\mathrm{mgVSS}/\mathrm{mgCOD})$$
where f_cv_ is a conversion factor of 1.48 mgCOD/mgVSS.

Step 3: µ_emp_ can be obtained from the relation between the biomass growth rate and the substrate degradation rate, as Equation (3) shows.
(3)$${µ}_{\mathrm{emp}}=\frac{Y_{H}{\cdot R}_{s}}{\left(1-Y_{H}{\cdot f}_{\mathrm{cv}}\right){\cdot X}_{H}}{\;(h}^{-1})$$

Step 4: By linearizing the Monod model, K_M_ and µ_m,H_ were assessed according to Equation (4).
(4)$$\frac{1}{{µ}_{\mathrm{emp}}}=\frac{1}{{µ}_{m,H}}+\frac{K_{M}}{{µ}_{m,H}}\cdot \frac{1}{S}\left(h\right)$$

Step 5: The estimation of b_H_ was performed according to Equation (5).
(5)$$b_{H}=\frac{{\mathrm{OUR}}_{\mathrm{end}}}{1{.42\cdot X}_{T}\cdot \left[1-Y_{H}\left(1-f_{p}\right)\right]\;}\left({\mathrm{day}}^{-1}\right)$$
where (1 − fp) is the fraction of volatile biomass (mgVSS/mgTSS) and OUR_end_ is the endogenous oxygen uptake rate.

Step 6: The assessment of kinetic parameters at working temperature was performed by applying the Metcalf equation [40], as shown in Equation (6).
(6)$$r_{T}{=r}_{20}{\cdot \theta }^{(T-20)}$$
where r_T_ and r_20_ are the kinetic parameters at working temperature and at 20 °C, respectively; θ is a fitting standard parameter with a value of 1.04 for the MBR; and T is the working temperature.

Step 7: The evaluation of the substrate degradation rate of organic matter removal, r_su,H_, was performed according to Equation (7).
(7)$$r_{\mathrm{su},H}=\frac{\mu_{m,H\cdot }{S\cdot X}_{H}}{Y_{H}\cdot \left(K_{M}+S\right)}\frac{{\mathrm{mgO}}_{2}}{L\cdot h}$$

Step 8: The estimation of net heterotrophic biomass growth rate, r’_x,H_, based on the biomass growth rate and the decay rate, was performed as shown in Equation (8).
(8)$$r_{x,H}^{\prime }=\frac{\mu_{m,H}\cdot S}{K_{M}+S\;}{\cdot X}_{H}-b_{H}{\cdot X}_{H}{\;(\mathrm{mgO}}_{2}/(L\cdot h))$$

## 3. Results and Discussion

### 3.1. Dynamic and Static Oxygen Uptake Rates

The exogenous respiration test results of operation phases 1 (12 h HRT) and 2 (18 h HRT) are included in Figure 4 and Figure 5, respectively.

In operation phase 1, the presence of BPA or ciprofloxacin did not modify the duration of the respirometric assays. On the other hand, the shock of carbamazepine increased the duration of the respirometric tests from 3000 to 3500 s. The mix of emerging pollutants showed the same effect as carbamazepine, extending the duration from 2600 to 3000 s approximately, which implies that there was more time for the heterotrophic biomass to degrade the organic matter substrate. This suggests a higher influence of carbamazepine on the duration of the kinetic experiments.

In relation to operation phase 2, both carbamazepine and ciprofloxacin caused an increase in the duration of the respirometric tests, around 500 s each. However, BPA did not change the duration of these experiments. The same trend was observed with the mix of emerging compounds, which suggests a higher effect of BPA over longer HRTs.

Regarding the values of R_s_, the pharmaceuticals reduced the three maximum values of R_s_ of the respirometric tests, with the exception of ciprofloxacin, in phase 1. This effect was not observed in phase 2, which could have been due to the fact that operation over a longer HRT compensated the possible reduction in R_s_. In light of this, the addition of ciprofloxacin even favoured the dynamic oxygen uptake rate.

In comparing the R_s_ values of both operation phases, it should be highlighted that phase 1 assays in the presence of pollutants had lower R_s_ values than phase 2 essays, which could have been due to the better operation conditions corresponding to second-phase samples, i.e., a longer HRT and slightly higher MLSS concentrations (Figure 4 and Figure 5).

The endogenous respiration tests of the eight pairs of experiments are included in Figure 6 and Figure 7.

In general, the presence of pollutants in both phases tended to decrease the maximum value of the OUR, i.e., the endogenous oxygen uptake rate (OUR_end_), compared with the values corresponding to the control tests. Additionally, as previously observed for R_s_, the OUR_end_ values in the presence of the emerging pollutants were higher in phase 2 than in phase 1, which might have been caused by the most favourable operation conditions of phase 2 (a longer HRT and higher MLSS concentrations).

The different trends of R_s_ and OUR were considered in order to evaluate the kinetic parameters following the mathematical procedure described in Equations (1)–(8).

### 3.2. Heterotrophic Kinetic Modelling

The kinetic parameters of heterotrophic biomass in the absence and presence of pollutants calculated following the kinetic modelling steps described in the Kinetic Modelling section are indicated in Table 4 and Table 5 at respirometry temperature (20 °C) and at operation temperature, respectively.

In addition, Figure 8 shows the variables r_su,H_ and r’_x,H_, which encompass the rest of the kinetic parameters, in the absence and presence of the emerging pollutants at 20 °C.

In addition, Figure 9 shows the variables r_su,H_ and r’_x,H_, which encompass the rest of the parameters, in the absence and presence of the emerging pollutants at operation temperature.

In phase 1, independently of the temperature, the maximum specific growth rate of heterotrophic biomass decreased by 62.26% and 49.82% in the presence of BPA and ciprofloxacin, respectively. However, this kinetic parameter increased by 21.32% when carbamazepine was added. In the presence of the mix of emerging pollutants, μ_m_ was higher (37.37%), which suggested a higher effect of carbamazepine. In phase 2, all contaminants reduced the μ_m_ values by 28.86% to 62.24%, with a reduction of 38.56% in the presence of the mix of emerging pollutants. Consequently, a longer HRT did not improve the maximum specific growth rate. Moreover, operation at low values of temperature (Table 3) decreased the values of the maximum specific growth rate by 37.54% to 44.47% in phase 1 and 42.25% to 44.47% in phase 2. The overall trend of reduction in μ_m_ implied that the heterotrophic biomass required more time to oxidize organic matter in the presence of emerging pollutants and at lower temperatures.

K_M_ showed a similar trend in both phases. In phase 1, we observed a reduction in the half-saturation coefficient for organic matter in the presence of BPA (69.99%) and ciprofloxacin (49.25%), with increase percentages of 27.16% and 127.49% with carbamazepine and the mix of emerging compounds, respectively. In phase 2, K_M_ was reduced in the presence of pollutants by 29.26% to 77.52% with the individual shocks and by 54.21% with the mix of pollutants. In this case, operation over a longer HRT seemed to improve the half-saturation coefficient of organic matter. The effect of the drop in temperature was identical to that obtained for the maximum specific growth rate with the same reduction percentages. The global pattern of the reduction in K_M_ could indicate that less available substrate was required to reach μ_m_, which indicates that the system was not inhibited by the substrate.

Regarding the amount of heterotrophic biomass produced per substrate oxidized, represented by the yield coefficient, the results do not show any clear trend, with increases ranging from 1.88% to 1.98% with BPA and from 0.56% to 1.55% with the mix of emerging pollutants in both phases. Carbamazepine exerted a negative effect on this parameter, with reduction percentages of 1.38–2.58%, whereas ciprofloxacin increased Y_H_ over the 12 h HRT (0.12%) and reduced it over the 18 h HRT (6.31%). Thus, the influence of the different emerging pollutants on this kinetic parameter was almost negligible. Nevertheless, a clear reduction in the amount of heterotrophic biomass produced per organic matter oxidized was observed at operation temperature, with reduction percentages similar to those obtained for μ_m_ and K_M_.

Finally, the b_H_ values decreased in the presence of pollutants by 24.76% to 66.51% with the individual shocks and by 15.19% with the mix of contaminants in phase 1. However, the trend changed in phase 2 with ciprofloxacin and the mix of emerging pollutants, with increases of 24.53% and 17.89%, respectively, which suggests a higher effect of the HRT with these shocks. Therefore, the biomass decay rate for heterotrophic biomass was reduced in the presence of pollutants with the exception of ciprofloxacin and the mix of emerging pollutants over the 18 h HRT. These latter two cases indicate a higher quantity of biomass being oxidized per day, which leads to higher loss of cell mass in the presence of ciprofloxacin, and its mix with BPA and carbamazepine over longer HRTs. As seen with μ_m_, the temperature also reduced the decay rate for heterotrophic bacteria, with reduction percentages similar to those observed at 20 °C.

The effect of these variations in the kinetic parameters that characterize the heterotrophic bacteria in the membrane bioreactor are included in the values of the substrate degradation rate of organic matter removal and the net heterotrophic biomass growth rate (Table 4 and Table 5). Both r_su,H_ and r’_x,H_ showed trends similar to those observed for μ_m_ and K_M_. In particular, in phase 1, r_su,H_ decreased by 61.27% and 62.03% in the presence of BPA at 20 °C and 5 °C, respectively, and lessened by 46.10% and 47.54% in the presence of ciprofloxacin at 20 °C and 7 °C, respectively. For its part, carbamazepine caused increases in r_su,H_ of 21.87% and 22.81% at 20 °C and 8 °C, respectively. The same pattern occurred with the mix of the three emerging pollutants, with increases in r_su,H_ of 31.23% and 33.41% at 20 °C and 6 °C, respectively, indicating a possible higher effect of carbamazepine in the mix of contaminants. In phase 2, over a longer HRT, all contaminants reduced the degradation rate of organic matter by 21.25% to 57.43% at 20 °C and 22.44% to 59.69% at 5–6 °C, with reductions of 35.94% at 20 °C and 37.47% at 5 °C in the presence of the mix of the emerging pollutants. On the other hand, no significant differences were observed when the temperature decreased from 20 °C to the operation values, with slight rises from 0.90% to 7.58%. It must be highlighted that independently of the temperature, organic matter was degraded faster over a longer HRT with similar doping due to the longer contact time between the substrate and microorganisms within the bioreactor.

Regarding the behaviour of r’_x,H_, the pattern was similar to that described for r_su,H_, with increases in r’_x,H_ of 58.25% and 61.08% at 20 °C and 6 °C in the presence of the mix of pollutants in phase 1. Nevertheless, in the presence of the mix of contaminants, over the 18 h HRT, the net heterotrophic biomass growth rate lessened by 43.84% and 45.19% at 20 °C and 5 °C, respectively. This could be based on lower biomass activity during operation over the 18 h HRT. Furthermore, operation at low values of temperature (Table 3) decreased the values of the net heterotrophic biomass growth rate by 35.03% to 43.66% in phase 1 and 37.18% to 42.77% in phase 2, which indicates an evident effect of temperature on r’_x,H_.

Concerning the synergic effect of the three compounds, it was not observed on r_su,H_ and r’_x,H_, independently of the temperature, since the values of these kinetic rates were included in the ranges obtained with the individual shock of each emerging pollutant. In particular, at operation temperature, the values of r_su,H_ with the individual additions varied from 18.6876 to 52.1435 mgO_2_/(L·h) over the 12 h HRT and changed from 45.9095 to 52.8674 mgO_2_/(L·h) over the 18 h HRT, where the values obtained with the mix of contaminants were 40.1192 and 46.6750 mgO_2_/(L·h), respectively. Regarding the values of r’_x,H_, these ranged from 4.0965 to 21.1980 mgVSS/(L·h) over the 12 h HRT and varied from 10.2832 to 14.5019 mgVSS/(L·h) over the 18 h HRT, where the values obtained with the mix of pollutants were 11.9389 and 12.3083 mgVSS/(L·h), respectively. In light of this, the three-compound synergic effect did not worsen the results obtained with the individual shocks.

It is important to point out that the behaviour of the heterotrophic biomass was contrary to that observed by other authors in the presence of BPA [28]. In particular, at temperatures of 12.1–31.1 °C, the above authors obtained r_su,H_ values varying from 50 to 200 mgO_2_/(L·h) and r’_x,H_ ranging from 20 to 170 mgVSS/(L·h) in the presence of BPA, which exceeded the values corresponding to the absence of this compound. In this study, the r_su,H_ values varied from 18.6876 to 45.9095 mgO_2_/(L·h), and r’_x,H_ ranged from 4.0965 to 14.5019 mgVSS/(L·h), which highlights a higher effect of very low temperatures on heterotrophic bacteria [35,36]. In light of this, [38] obtained that the presence of a mix of carbamazepine, ciprofloxacin, and ibuprofen almost doubled the r_su,H_ of the control experiment in a membrane bioreactor, with values fluctuating from 183.97 to 192.88 mgO_2_/(L·h) at temperatures varying from 18.9 to 21.4 °C, and the same trend was obtained by [41] in a membrane bioreactor at 12.6 and 21.5 °C. The authors of [29] did not observe any inhibition of COD removal under exogenous respiration in the presence of carbamazepine at 20 °C and similar MLSS and HRT conditions. This strengthens the higher influence of very low temperatures on the kinetic behaviour of heterotrophic bacteria, which could hinder the acclimatization of bacterial populations to the presence of micropollutants.

## 4. Conclusions

Based on the kinetic results in the presence and absence of micropollutants (bisphenol A, carbamazepine, ciprofloxacin, and the mixture of the three above compounds) obtained with respirometric tests carried out on heterotrophic biomass in a membrane bioreactor working over two different hydraulic retention times (12 and 18 h), at low-temperature conditions (5–8 °C), and at low MLSS concentrations (1633–2866 mg/L), the following conclusions were drawn:In general, the shocks of high-concentration bisphenol A, carbamazepine, and ciprofloxacin worsened the degradation rate of organic matter and the net heterotrophic biomass growth rate, with the exception of carbamazepine and the mix of micropollutants over the 12 h HRT, which caused the reactivation of biomass with values of 52.1435 mgO_2_/(L·h) and 21.1980 mgVSS/(L·h) with carbamazepine at 8 °C, and 40.1192 mgO_2_/(L·h) and 11.9389 mgVSS/(L·h) with the mix of emerging pollutants at 6 °C. This could have been due to higher biomass activity over a lower HRT.The three-compound synergic effect did not worsen the biomass performance compared with the effect of single-compound shocks, independently of temperature, with values of 40.1192 mgO_2_/(L·h) with the mix in relation to the range 18.6876–52.1435 mgO_2_/(L·h) with the individual shocks over the 12 h HRT and at operation temperature. Over the 18 h HRT, the values of single doping ranging from 45.9095 to 52.8674 mgO_2_/(L·h) also encompassed the value of the mix (46.6750 mgO_2_/(L·h)) at operation temperature. The trend of r’_x,H_ was similar to that observed for r_su,H_.Low temperatures mainly inhibited the net heterotrophic biomass growth rate, with reductions from 19.8848 to 11.9389 mgVSS/(L·h) and from 21.5082 to 12.3083 mgVSS/(L·h) in the presence of the mix of emerging pollutants over the 12 h and 18 h HRTs, respectively.The hydraulic retention time was the most influential variable regarding the degradation rate of organic matter, independently of temperature, with faster biodegradation rates over 18 h compared with 12 h with similar doping. Thus, hydraulic retention time plays an important role in biomass performance even at low temperatures, allowing membrane bioreactors to work under such extreme temperature conditions under the shock of micropollutants by increasing its value.

## Figures and Tables

**Figure 1 membranes-13-00419-f001:**
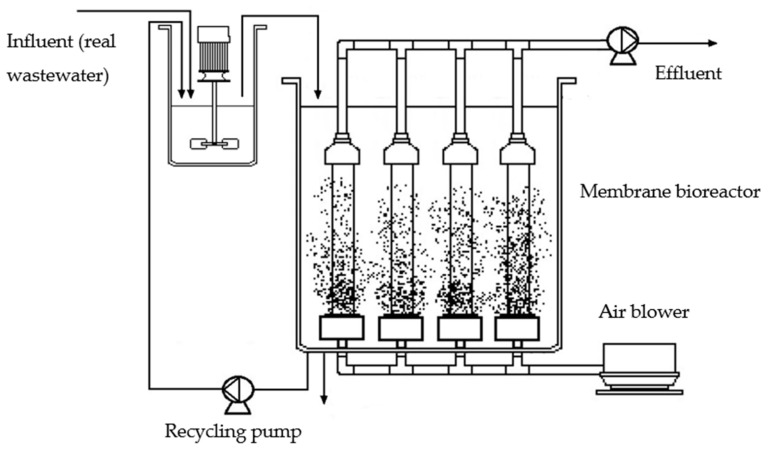
Schematic diagram of the pilot plant.

**Figure 2 membranes-13-00419-f002:**
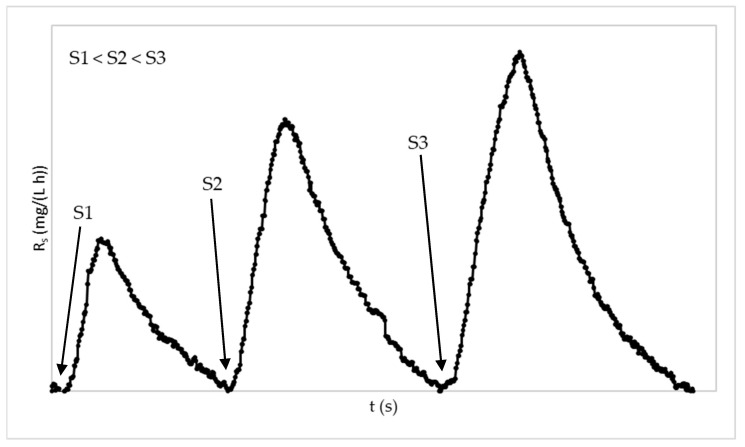
Evolution example of dynamic oxygen uptake rate (R_s_) in exogenous respirometric experiment and addition times of the three different dilutions of sodium acetate substrate. S1: sodium acetate addition of 5 mL; S2: sodium acetate addition of 10 mL; S3: sodium acetate addition of 15 mL.

**Figure 3 membranes-13-00419-f003:**
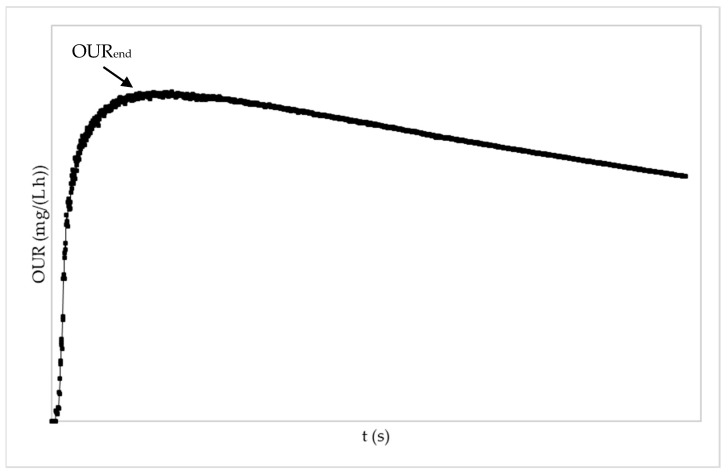
Evolution example of static oxygen uptake rate (OUR) in endogenous respirometric assay. OUR_end_: maximum rate of consumption in endogenous respiration assay.

**Figure 4 membranes-13-00419-f004:**
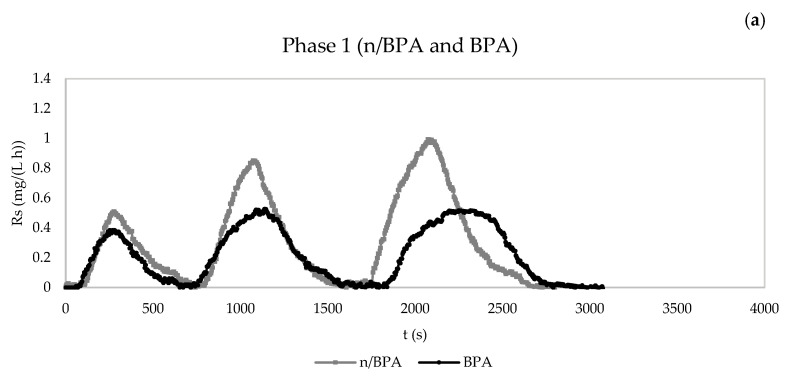
Evolution of dynamic oxygen uptake rate (R_s_) in the respirometric experiments in the absence and presence of pollutants for the determination of the kinetic parameters. (**a**) Phase 1: control without BPA and with BPA. (**b**) Phase 1: control without carbamazepine and with carbamazepine. (**c**) Phase 1: control without ciprofloxacin and with ciprofloxacin. (**d**) Phase 1: control without pharmaceuticals and with mixture of pharmaceuticals. n/BPA: absence of bisphenol A. BPA: presence of bisphenol A. n/CBZ: absence of carbamazepine. CBZ: presence of carbamazepine. n/CPF: absence of ciprofloxacin. CPF: presence of ciprofloxacin. n/Mix: absence of mix of pharmaceuticals. Mix: presence of mix of pharmaceuticals.

**Figure 5 membranes-13-00419-f005:**
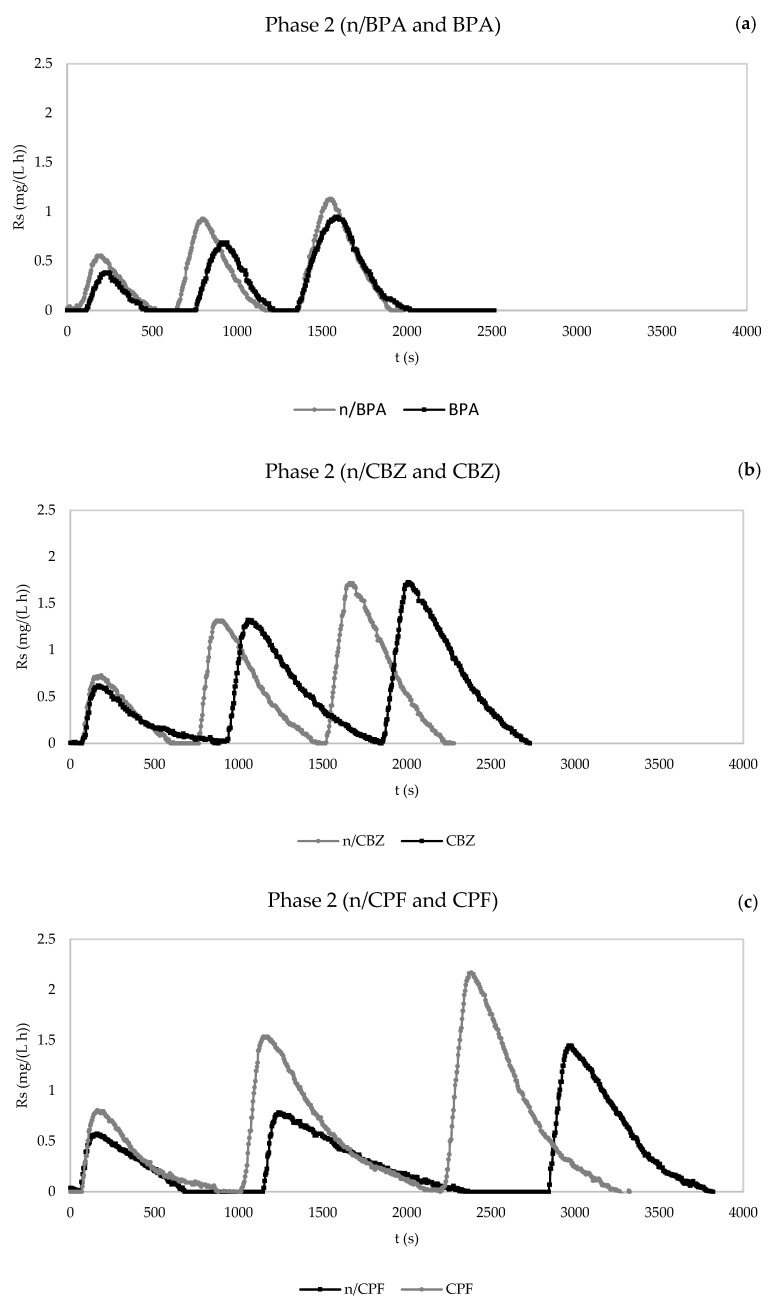
Evolution of dynamic oxygen uptake rate (R_s_) in the respirometric experiments in the absence and presence of pollutants for the determination of the kinetic parameters. (**a**) Phase 2: control without BPA and with BPA. (**b**) Phase 2: control without carbamazepine and with carbamazepine. (**c**) Phase 2: control without ciprofloxacin and with ciprofloxacin. (**d**) Phase 2: control without pharmaceuticals and with mixture of pharmaceuticals. n/BPA: absence of bisphenol A. BPA: presence of bisphenol A. n/CBZ: absence of carbamazepine. CBZ: presence of carbamazepine. n/CPF: absence of ciprofloxacin. CPF: presence of ciprofloxacin. n/Mix: absence of mix of pharmaceuticals. Mix: presence of mix of pharmaceuticals.

**Figure 6 membranes-13-00419-f006:**
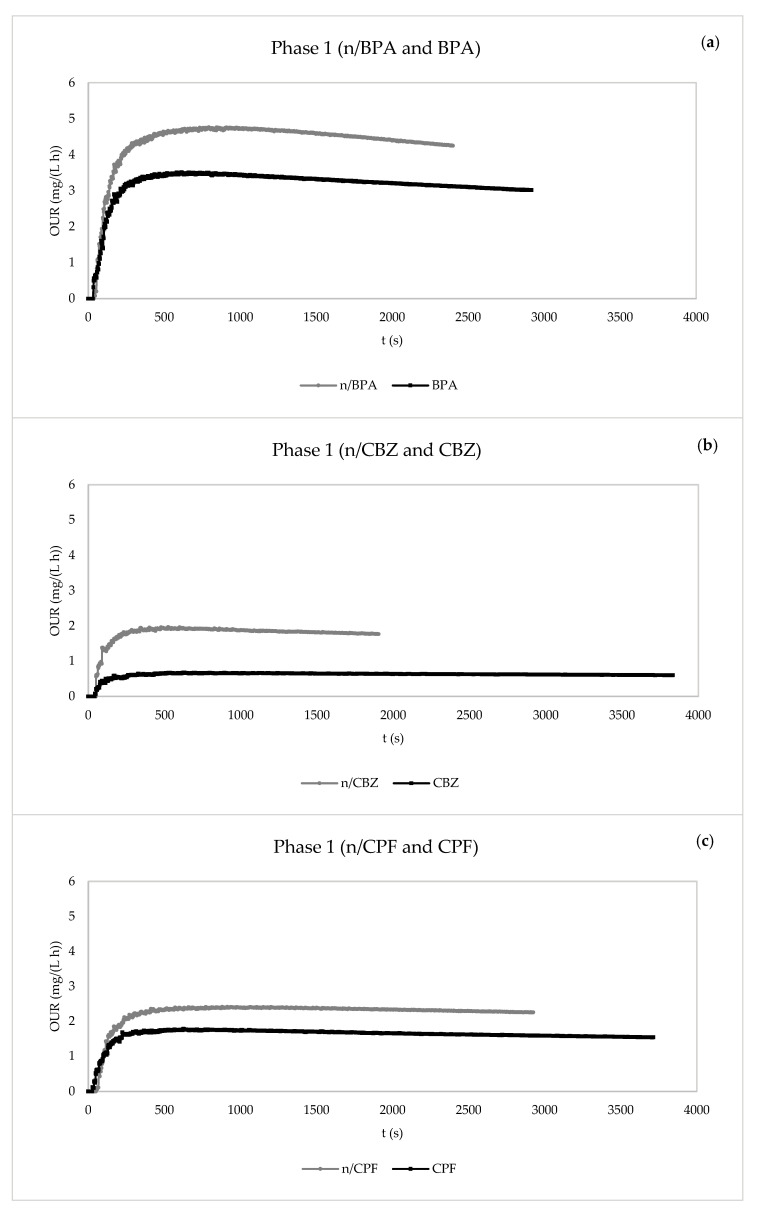
Evolution of static oxygen uptake rate (OUR) in the respirometric experiments in the absence and presence of pollutants for the determination of the kinetic parameters. (**a**) Phase 1: control without BPA and with BPA. (**b**) Phase 1: control without carbamazepine and with carbamazepine. (**c**) Phase 1: control without ciprofloxacin and with ciprofloxacin. (**d**) Phase 1: control without pharmaceuticals and with mixture of pharmaceuticals. n/BPA: absence of bisphenol A. BPA: presence of bisphenol A. n/CBZ: absence of carbamazepine. CBZ: presence of carbamazepine. n/CPF: Absence of ciprofloxacin. CPF: Presence of ciprofloxacin. n/Mix: absence of mix of pharmaceuticals. Mix: presence of mix of pharmaceuticals.

**Figure 7 membranes-13-00419-f007:**
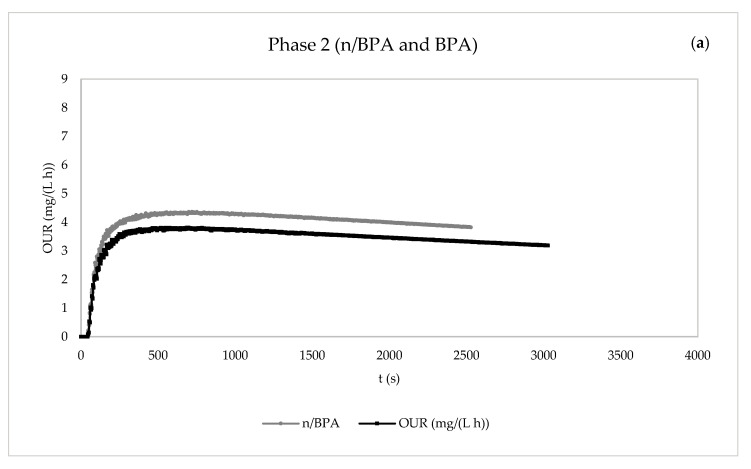
Evolution of static oxygen uptake rate (OUR) in the respirometric experiments in the absence and presence of pollutants for the determination of the kinetic parameters. (**a**) Phase 2: control without BPA and with BPA. (**b**) Phase 2: control without carbamazepine and with carbamazepine. (**c**) Phase 2: control without ciprofloxacin and with ciprofloxacin. (**d**) Phase 2: control without pharmaceuticals and with mixture of pharmaceuticals. n/BPA: absence of bisphenol A. BPA: presence of bisphenol A. n/CBZ: absence of carbamazepine. CBZ: presence of carbamazepine. n/CPF: absence of ciprofloxacin. CPF: presence of ciprofloxacin. n/Mix: absence of mix of pharmaceuticals. Mix: presence of mix of pharmaceuticals.

**Figure 8 membranes-13-00419-f008:**
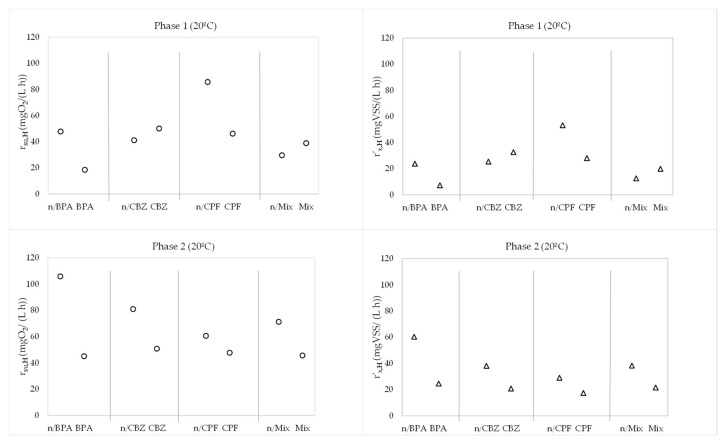
Evolution of r_su,H_ (◦) and r_’x,H (_Δ) in phase 1 and phase 2 at 20 °C. n/BPA: absence of bisphenol A. BPA: presence of bisphenol A. n/CBZ: absence of carbamazepine. CBZ: presence of carbamazepine. n/CPF: absence of ciprofloxacin. CPF: presence of ciprofloxacin. n/Mix: absence of mix of pharmaceuticals. Mix: presence of mix of pharmaceuticals.

**Figure 9 membranes-13-00419-f009:**
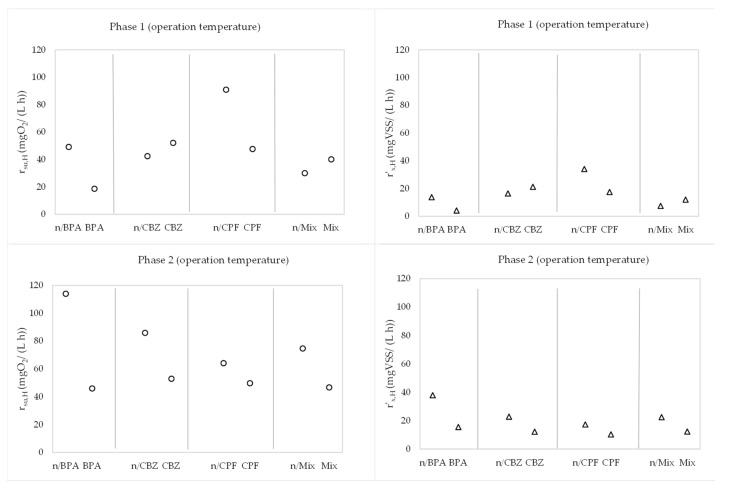
Evolution of r_su,H_ (◦) and r_’x,H_ (Δ) in phase 1 and phase 2 at operation temperature in plant. n/BPA: absence of bisphenol A. BPA: presence of bisphenol A. n/CBZ: absence of carbamazepine. CBZ: presence of carbamazepine. n/CPF: absence of ciprofloxacin. CPF: presence of ciprofloxacin. n/Mix: absence of mix of pharmaceuticals. Mix: presence of mix of pharmaceuticals.

**Table 1 membranes-13-00419-t001:** Permeability working values, where TMP is the transmembrane pressure.

			Pressure (bar)	∆TMP (bar)	Permeability (m^3^/(m^2^ h bar))	Permeability (L/(m^2^ h bar))
12 HRT	December	Suction	0.05	0.95	0.001986	1.99
		Backwashing	0.15	0.85	0.002190	2.22
	December	Suction	0.20	0.80	0.002358	2.36
		Backwashing	0.15	0.85	0.002219	2.22
18 HRT	January	Suction	0.10	0.90	0.001395	1.39
		Backwashing	0.25	0.75	0.001674	1.67
	January	Suction	0.03	0.97	0.001287	1.29
		Backwashing	0.25	0.75	0.001674	1.67

**Table 2 membranes-13-00419-t002:** Water characterization of phase-one and -two samples, where SSs—suspended solids; VSSs—volatile suspended solids; MLSSs—mixed liquor suspended solids; MLVSSs—mixed liquor volatile suspended solids; COD—chemical oxygen demand; and BOD_5_—five-day biochemical oxygen demand.

Respirometry	Sample	SSs (mg/L)	VSSs (mg/L)	pH	Conductivity (μS/cm)	COD (mgO_2_/L)	BOD_5_ (mgO_2_/L)
1	Influent	70	62	7.65	1087	448	240
2	Influent	28	4	7.94	1235	465	300
3	Influent	80	59	7.50	1122	388	230
4	Influent	94	75	7.63	1274	N/D*	310
5	Influent	78	75	7.68	1178	441	280
6	Influent	77	64	7.48	1020	404	310
7	Influent	49	38	7.80	1030	N/D*	310
8	Influent	109	91	7.66	1178	461	320
1	Effluent	2	1	7.74	974	54	36
2	Effluent	1	Non detected	7.34	886	46	29
3	Effluent	9	4	7.04	886	47	13
4	Effluent	10	1	6.98	918	47	13
5	Effluent	13	11	7.58	1149	46	9
6	Effluent	11	8	8.06	1045	47	36
7	Effluent	4	Non detected	8.23	1078	46	44
8	Effluent	8	3	8.10	1085	48	19

**Table 3 membranes-13-00419-t003:** Design of experiments.

Respirometric Assay	Emerging Compound	MLSSs (mg/L)	HRT (h)	F/M (kg BOD_5_/(kg MLSS day))	T (°C)	pH	Conductivity (μS/cm)
1	Bisphenol A	2033	12	0.010	5	7.07	895
2	Carbamazepine	1633	12	0.015	8	7.11	898
3	Ciprofloxacin	2333	12	0.008	7	7.50	880
4	Mixture	2166	12	0.012	6	7.00	908
5	Bisphenol A	2400	18	0.006	6	8.31	1055
6	Carbamazepine	2333	18	0.007	5	7.92	1058
7	Ciprofloxacin	2000	18	0.009	5	7.85	1083
8	Mixture	2866	18	0.006	5	7.98	973

**Table 4 membranes-13-00419-t004:** Kinetic parameters obtained by Equations (1)–(8) for phase one (1) and two (2) of experiments at respirometry temperature of 20 °C (where Mix—the mixture of the three compounds; μ_m_—the maximum specific growth rate of heterotrophic biomass; K_M_—the half-saturation coefficient of organic matter; Y_H,O2_—the yield coefficient for heterotrophic biomass referred to oxygen; b_H_—the decay coefficient of heterotrophic biomass; r_su,H_—the substrate degradation rate of organic matter removal; and r’_x,H_—the net heterotrophic biomass growth rate).

Emerging Pollutant	Condition	μ_m_ (h^−1^)	K_M_ (mgO_2_/L)	Y_H_ (mgVSS/mgO_2_)	b_H_ (day^−1^)	r_su,H_ (mgO_2_/(L h))	r’_x,H_ (mgVSS/(L h))
BPA (1)	Control	0.0224	3.6823	0.6243	0.1011	47.8169	23.8453
	Doped	0.0084	1.1049	0.6367	0.0761	18.5209	7.27150
Carbamazepine (1)	Control	0.0259	4.0224	0.6868	0.0573	41.1775	25.4455
	Doped	0.0314	5.1148	0.6691	0.0192	50.1847	32.6281
Ciprofloxacin (1)	Control	0.0378	7.8164	0.6616	0.0478	85.7522	53.2429
	Doped	0.0189	3.9671	0.6624	0.0351	46.2222	28.0582
Mixture (1)	Control	0.0136	1.5602	0.6597	0.1132	29.6647	12.5654
	Doped	0.0187	3.5492	0.6634	0.0960	38.9290	19.8848
BPA (2)	Control	0.0455	9.2084	0.6597	0.1240	105.8741	60.3232
	Doped	0.0172	2.0704	0.6720	0.0748	45.0740	24.5512
Carbamazepine (2)	Control	0.0290	6.8468	0.6031	0.1336	80.9733	38.0842
	Doped	0.0172	4.5470	0.5948	0.1172	50.7934	20.7853
Ciprofloxacin (2)	Control	0.0283	6.3899	0.6083	0.1273	60.6133	28.9932
	Doped	0.0201	4.5200	0.5699	0.1585	47.7319	17.3922
Mixture (2)	Control	0.0215	5.4844	0.6281	0.0668	71.2456	38.3015
	Doped	0.0132	2.5114	0.6378	0.0788	45.6430	21.5082

(1): phase 1 (12 h HRT); (2): phase 2 (18 h HRT).

**Table 5 membranes-13-00419-t005:** Kinetic parameters obtained using Equations (1)–(8) for phase one (1) and two (2) of experiments at operation temperature existing in plant (where Mix—the mixture of the three compounds; μ_m_—the maximum specific growth rate of heterotrophic biomass; K_M_—the half-saturation coefficient of organic matter; Y_H,O2_—the yield coefficient of heterotrophic biomass referred to oxygen; b_H_—the decay coefficient of heterotrophic biomass; r_su,H_—the substrate degradation rate of organic matter removal; and r’_x,H_—the net heterotrophic biomass growth rate).

Emerging Pollutant	Condition	μ_m_ (h^−1^)	K_M_ (mgO_2_/L)	Y_H_ (mgVSS/mgO_2_)	b_H_ (day^−1^)	r_su,H_ (mgO_2_/(L h))	r’_x,H_ (mgVSS/(L h))
BPA (1)	Control	0.0124	2.0447	0.3467	0.0562	49.2141	13.7248
	Doped	0.0047	0.6135	0.3535	0.0423	18.6876	4.0965
Carbamazepine (1)	Control	0.0162	2.5124	0.4290	0.0358	42.4592	16.4430
	Doped	0.0196	3.1947	0.4179	0.0120	52.1435	21.1980
Ciprofloxacin (1)	Control	0.0227	4.6943	0.3974	0.0287	90.9312	34.0342
	Doped	0.0114	2.3825	0.3978	0.0211	47.7053	17.4411
Mixture (1)	Control	0.0079	0.9010	0.3809	0.0653	30.0730	7.4117
	Doped	0.0108	2.0496	0.3831	0.0554	40.1192	11.9389
BPA (2)	Control	0.0263	5.3176	0.3809	0.0716	113.9012	37.8930
	Doped	0.0099	1.1956	0.3881	0.0432	45.9095	15.5019
Carbamazepine (2)	Control	0.0161	3.8018	0.3349	0.0742	85.8267	22.7722
	Doped	0.0095	2.5248	0.3303	0.0651	52.8674	12.2263
Ciprofloxacin (2)	Control	0.0157	3.5481	0.3378	0.0707	64.0897	17.2732
	Doped	0.0112	2.5098	0.3164	0.0880	49.7099	10.2832
Mixture (2)	Control	0.0119	3.0453	0.3488	0.0371	74.6500	22.4548
	Doped	0.0073	1.3945	0.3542	0.0437	46.6750	12.3083

(1): phase 1 (12 h HRT); (2): phase 2 (18 h HRT).

## Data Availability

Not applicable.

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
