# Peer review of "Kinetic Effects of Ciprofloxacin, Carbamazepine, and Bisphenol on Biomass in Membrane Bioreactor System at Low Temperatures to Treat Urban Wastewater"

_membranes, 2023, doi:10.3390/membranes13040419_

Round 1

Reviewer 1 Report

line 103-107, seemed to be reduplicative aims comparing with previous paragraph.

table 2, SS/VSS but not MLSS/MLVSS are used as water quality paramter; MLSS/MLVSS are usually paramters used in biological treatment tank (e.g. MLSS in aeration tank).

From table 2, it is quite weird that SS of effluent from 0.04um uf membrane was higher than 5 mg/L.

Table 3, format issue

Fig. 4,5,6, and 7, it would be better to put control and doped data together into one graph for comparing.

Author Response

Manuscript ID Number: Membrane - 2229380

Manuscript Title: Kinetic effect of ciprofloxacin, carbamazepine, and bisphenol on the biomass in membrane bioreactor system at low temperature to treat urban wastewater.

Article Type: Article

Reviewer comments

Reviewer #1:

Thank you very much for your revision and your considerations about the paper. We hope the content of this manuscript has been improved after the revision with your comments. We will answer each of them.

Improvements:

  1. Line 103-107, seemed to be reduplicative aims comparing with previous paragraph:

Answer: Thank you so much for your consideration. To avoid this, the following text before the main objectives has been removed: ‘In light of this, the present study aims to check the effect of low temperature conditions in the performance of the heterotrophic biomass of an MBR system under the presence of the three compounds cited before, carbamazepine, ciprofloxacin, and BPA, evaluating the kinetics by respirometric assays.’

  1. Table 2, SS/VSS but not MLSS/MLVSS are used as water quality paramter; MLSS/MLVSS are usually parameters used in biological treatment tank (e.g. MLSS in aeration tank).

Answer: Thank you so much for this annotation. This was an error in notation and it has been corrected.

  1. From table 2, it is quite weird that SS of effluent from 0.04 um uf membrane was higher than 5 mg/L.

Answer: Thank you so much for this comment. This was detected during the study and possibly these values could be due the backwash tank of the membranes during the different experimental phases. The following text it has been included in Section 2.1 ‘In the effluent, values of suspended solids surpassing 5 mg/L were detected in some cases, which may be due to the membrane backwash from the effluent tank during the different experimental phases.’

  1. Table 3, format issue

Answer: Thank you so much for this observation. The formatting issue in Table 3 has been corrected.

  1. 4,5,6, and 7, it would be better to put control and doped data together into one graph for comparing:

Answer: Thank you so much for this annotation. Figures 4, 5, 6 and 7 have been modified to put control and doped data together into one graph for comparing.  

Reviewer 2 Report

This study analysed the kinetic results in presence and absence of micropollutants (bisphenol A, carbamazepine, ciprofloxacin, and the mixture of the three compounds) obtained from respirometric tests to the mixed liquor for the heterotrophic biomass in a membrane bioreactor (MBR) working at two different hydraulic retention times (12 and 18 h) and low temperature conditions (5-8⁰C). The experimental data seem carefully analyzed and explained. However, the following points need to be revised:

1.     In the abstract, it’s mentioned:

“Temperature played a major role due to its inhibition effect, mainly affecting the net heterotrophic biomass growth rate in the presence of pharmaceuticals for both hydraulic retention time (12 and 18 h of HRT). The hydraulic retention time of the system was the most influential variable in the system affecting the degradation rate of organic matter, with the greatest effect at 18 h of HRT.”

These statements are confusing. Which variable is more important: temperature or hydraulic retention time (HRT)? The authors should revise the statements. It seems to me that the inhibition effect, mainly affecting the net heterotrophic biomass growth rate in the presence of pharmaceuticals for both hydraulic retention time (12 and 18 h of HRT), is at low temperatures (5-8⁰C). But low temperature condition for the inhibition effect is not mentioned in the statement.

2.     Table 4 is for experiments at respirometry temperature of 20ºC and Table 5 is for experiments at operation temperature (5-8⁰C) existing in plant. The influence of temperature is discussed in the discussion after Tables 4-5. However, the influence of temperature is not well mentioned in Conclusions.

In Conclusions, it is only mentioned “Low temperature mainly inhibited the net heterotrophic biomass growth rate with values of 11.9389 and 12.3083 mgVSS/(L·h) in presence of the mix of emerging pollutants at 12 h and 18 h of HRT, respectively.” The statement should be revised to make the influence of temperature more clear.

3.     Some errors in the text, such as:

On line 98, “with mayor influence” should be “with major influence”.

On line 209, “In order to the kinetic modelling of the analysed biomass” should be “In order to + verb”.

On line 140, BOD5 should be BOD subscript 5.

On line 222, “Step 5: estimation of …..” should be “Step 5: Estimation of ….”.

Please have someone to check the text.

Author Response

Manuscript ID Number: Membrane - 2229380

Manuscript Title: Kinetic effect of ciprofloxacin, carbamazepine, and bisphenol on the biomass in membrane bioreactor system at low temperature to treat urban wastewater.

Article Type: Article

Reviewer comments

Reviewer #2:

This study analysed the kinetic results in presence and absence of micropollutants (bisphenol A, carbamazepine, ciprofloxacin, and the mixture of the three compounds) obtained from respirometric tests to the mixed liquor for the heterotrophic biomass in a membrane bioreactor (MBR) working at two different hydraulic retention times (12 and 18 h) and low temperature conditions (5-8⁰C). The experimental data seem carefully analyzed and explained. However, the following points need to be revised.

Thank you very much for your revision and your considerations about the paper. We hope the content of this manuscript has been improved after the revision with your comments. We will answer each of them.

Improvements:

  1. In the abstract, it’s mentioned:

‘Temperature played a major role due to its inhibition effect, mainly affecting the net heterotrophic biomass growth rate in the presence of pharmaceuticals for both hydraulic retention time (12 and 18 h of HRT). The hydraulic retention time of the system was the most influential variable in the system affecting the degradation rate of organic matter, with the greatest effect at 18 h of HRT.’

These statements are confusing. Which variable is more important: temperature or hydraulic retention time (HRT)? The authors should revise the statements. It seems to me that the inhibition effect, mainly affecting the net heterotrophic biomass growth rate in the presence of pharmaceuticals for both hydraulic retention time (12 and 18 h of HRT), is at low temperatures (5-8⁰C). But low temperature condition for the inhibition effect is not mentioned in the statement.

Answer: Thank you so much for your consideration. In the abstract the mentioned text has been replaced by ‘Independent of temperature, organic substrate was biodegraded faster at higher hydraulic retention time (HRT) for similar dopings, which was probably due to the higher contact time between substrate and microorganisms within the bioreactor. However, low values of temperature negatively affected the net heterotrophic biomass growth rate with reductions from 35.03 to 43.66% in phase 1 (12 h of HRT), and from 37.18 to 42.77% in phase 2 (18 h of HRT).’

  1. Table 4 is for experiments at respirometry temperature of 20ºC and Table 5 is for experiments at operation temperature (5-8⁰C) existing in plant. The influence of temperature is discussed in the discussion after Tables 4-5. However, the influence of temperature is not well mentioned in Conclusions.

In Conclusions, it is only mentioned ‘Low temperature mainly inhibited the net heterotrophic biomass growth rate with values of 11.9389 and 12.3083 mgVSS/(L·h) in presence of the mix of emerging pollutants at 12 h and 18 h of HRT, respectively.’ The statement should be revised to make the influence of temperature more clear.

Answer: Thank you so much for this annotation. In conclusions the mentioned text has been changed by ‘Low temperature mainly inhibited the net heterotrophic biomass growth rate with reductions from 19.8848 to 11.9389 mgVSS/(L·h) and from 21.5082 to 12.3083 mgVSS/(L·h) in presence of the mix of emerging pollutants at 12 h and 18 h of HRT, respectively.’

  1. Some error in the text, such as:
  • On line 98, ‘with mayor influence’ should be ‘with major influence’.
  • On line 209, ‘In order to kinetic modelling of the analysed biomass’ should be ‘In order to + verb’.
  • On line 140, BOD5 should be BOD subscript 5.
  • On line 222, ‘Step 5: estimation of…..’ should be ‘Step 5: Estimation of …..’.

Answer: Thank you so much for these observations. The text has been corrected in its annotations and has been revised.  

Reviewer 3 Report

In this paper, after sampling sludge from MBR reactors with different retention times, the effect of trace pollutants and temperature on the kinetic coefficient was confirmed by measuring the respiration rate. Although this manuscript was written based on sufficient prior research, it seems that the following modifications are needed.

1. Compared to the general activated sludge process, MBR has a longer SRT, and changes in retention time cause changes in F/M. However, there is no mention related to this.

2. For the reader's understanding, it is necessary to explain the difference between dynamic oxygen uptake rate(Rs) and oxygen uptake rate(OUR). It is also necessary to explain the meaning of the values ​​and trends in the graph.

3. It would be good if the results were graphed so that the explanations related to Tables 4 and 5 could stand out.

Author Response

Manuscript ID Number: Membrane - 2229380

Manuscript Title: Kinetic effect of ciprofloxacin, carbamazepine, and bisphenol on the biomass in membrane bioreactor system at low temperature to treat urban wastewater.

Article Type: Article

Reviewer comments

Reviewer #3:

In this paper, after sampling sludge from MBR reactors with different retention times, the effect of trace pollutants and temperature on the kinetic coefficient was confirmed by measuring the respiration rate. Although this manuscript was written based on sufficient prior research, it seems that the following modifications are needed.

Thank you very much for your revision and your considerations about the paper. We hope the content of this manuscript has been improved after the revision with your comments. We will answer each of them.

Improvements:

  1. Compared to the general activated sludge process, MBR has a longer SRT, and changes in retention time cause changes in F/M. However, there is no mention related to this.

Answer: Thank you so much for this observation. The following column has been included in Table 3 in Section 2.3.

Respirometry assays

Emerging compound

MLSS (mg/L)

HRT (h)

F/M

(kg BOD5/(kg MLSS day))

T

(ºC)

pH

Conductivity (μS/cm)

1

Bisphenol A

2033

12

0.010

5

7.07

895

2

Carbamazepine

1633

12

0.015

8

7.11

898

3

Ciprofloxacin

2333

12

0.008

7

7.50

880

4

Mixture

2166

12

0.012

6

7.00

908

5

Bisphenol A

2400

18

0.006

6

8.31

1055

6

Carbamazepine

2333

18

0.007

5

7.92

1058

7

Ciprofloxacin

2000

18

0.009

5

7.85

1083

8

Mixture

2866

18

5

7.98

973

The following text has been added in Section 2.3 ‘It can be seen that under the operating conditions, the F/M ratios correspond to low pollutant loads with respect to the concentration of microorganisms to compensate for the low temperature effect.’

  1. For the reader's understanding, it is necessary to explain the difference between dynamic oxygen uptake rate (Rs) and oxygen uptake rate (OUR). It is also necessary to explain the meaning of the values ​​and trends in the graph.

Answer: Thank you so much for this comment. The following text has been added in Figure 2 ‘Evolution example of dynamic oxygen uptake rate (Rs) for exogenous respirometric experiment and addition time of the three different dilutions of sodium acetate substrate. S1: Sodium acetate addition of 5 mL; S2: Sodium acetate addition of 10 mL; S3: Sodium acetate addition of 15 mL.’

The following text has been included in Figure 3 ‘OURend: Máximum rate of consumption in endogenous respiration assay’.

The following text has been included in Section 2.3 ‘The Rs rate is the rate of consumption in exogenous assays with the presence of substrate and a constant supply of O2, while the OUR rate is the rate of consumption in the absence of substrate and without O2 supply.’

  1. It would be good if the results were graphed so that the explanations related to Tables 4 and 5 could stand out.

Answer: Thank you so much for this annotation. Figures 8 and 9 included in the document represent the values in tables 4 and 5. The following text has been included in Section 3.2 ‘In addition, Figure 8 shows the variables rsu,H and r'x,H, which encompass the rest of the kinetic parameters, in absence and presence of emerging pollutants at 20ºC’.

The following text has been included in Section 3.2 ‘In addition, Figure 9 shows the variables rsu,H and r'x,H, which encompass the rest of the parameters, in absence and presence of emerging pollutants at operation temperature.’
